# A Drop-In Solution for On-the-Fly Adaptation of Speculative Decoding in Large Language Models

## Abstract

Large Language Models (LLMs) are cutting-edge generative AI models built on transformer architecture, which tend to be highly memory-intensive when performing real-time inference. Various strategies have been developed to enhance the end-to-end inference speed for LLMs, one of which is speculative decoding. This technique involves running a smaller LLM (draft model) for inference over a defined window size, denoted as $\gamma$, while simultaneously being validated by the larger LLM (target model). Choosing the optimal $\gamma$ value and the draft model is essential for unlocking the potential of speculative decoding. But it is difficult to do due to the complicated influence from various factors, including the nature of the task, the hardware in use, and the combination of the large and small models. This paper introduces *on-the-fly adaption of speculative decoding*, a solution that dynamically adapts the choices to maximize the efficiency of speculative decoding for LLM inferences. As a drop-in solution, it needs no offline benchmarking or training. Experiments show that the solution can lead to 3.55-16.48% speed improvement over the standard speculative decoding, and 1.2-3.4$\times$ over the default LLMs.

## 1 Introduction

Large Language Models (LLMs) are state-of-the-art generative AI models built on transformer-based blocks (Brown et al., 2020; Ouyang et al., 2022). LLMs have an enormous number of parameters, and recent research not only focuses on training them efficiently but also explores how to optimize inference performance. In fact, there is evidence indicating that even small improvement in LLM inference speeds can result in significant cost savings. For instance, Google's infrastructure optimizations have demonstrated that improving inference efficiency can lead to substantial reductions in operational expenses. In large-scale deployments, a 1% increase in speed can indeed translate into millions of dollars saved (AI, 2023; Cloud, 2023).

Due to the autoregressive and memory-intensive nature of LLMs, it is challenging to optimize its inference throughput. Sampling for a new token depends on the previously generated tokens. Researchers are exploring mainly two approaches to circumvent this sequential dependence for more efficient parallel executions. One is to change the model architecture thus sampling granularity to parallelize the decoding process. Medusa (Cai et al., 2024), for example, introduces multiple decoding heads to generate tokens in parallel; Lookahead Decoding (Jacobi Decoding) (Fu et al., 2024) generates multiple tokens in parallel using nonlinear systems. This approach changes the neural architecture and hence requires new training, the high costs of which makes them difficult to adopt in practice. The other approach is speculative decoding (Leviathan et al., 2023; Chen et al., 2023). This approach first runs inference with a smaller LLM $M_q$, called the *draft model*, to generate the next $\gamma$ tokens ($\gamma$ is called speculation *window size*). After generating one window of tokens (called a *speculation step*), a *verification step* uses the Large LLM $M_p$, called the *target model*, to validate those tokens *in parallel*. Upon finding the first incorrect token, the execution throws away the rest of the tokens speculated by the draft model in that window and corrects the first rejected token (or appends a new token when all of the tokens are accepted). From there, it continues the speculation-validation process. This approach allows direct use of the pretrained LLMs, making it easier for adoption.

What is crucial for unlocking the potential of speculative decoding is to choose the best speculation window length, $\gamma$, and the best draft model to use. The best choices depend on the nature of the inference task, target model, software stack, hardware, and resource availability or workload changes (if running in a cloud). Suboptimal choices may not only substantially throttle the benefits but sometimes cause slowdowns to the inference (see Section 6). The standard approach (Leviathan et al., 2023; Chen et al., 2023) relies on offline trial-and-error-based search, which not only takes long time, but more importantly, cannot adapt to the changes in the tasks, target models, software stacks, hardware or other runtime changes. A recent study, SpecDec++ (Huang et al., 2024), attempts to improve it through a machine learning model. It trains a ResNet with many samples in offline data collection, and uses it to predict, at each generated token in actual inferences, whether the execution should stop speculation, so as to adapt the speculation window. Although the work shows some improvement in experiments, it requires hundreds or thousands of GPU-hours (Section 6.2) to train the model for one target-draft pair on one kind of task and one software and hardware configuration. Modern LLM servers often host many LLMs and their variants (e.g., different quantizations, with Lora or other fine-tuning models) and have various software and hardware configurations and task types, making the solution difficult to adopt in production systems.

This paper describes the first-known exploration of *on-the-fly adaptive speculation*, a drop-in solution that adapts speculative decoding at runtime without ahead-of-time training. Our exploration covers both speculation window size $\gamma$ and the choice of draft models. It experiments with several agile online methods for the adaptation, including a state machine-based mechanism, a cache-enabled state machine-based method, a reinforcement learning-based approach, and a token accuracy-based online window size optimization method. It analyzes these methods and evaluates them on four LLMs across three GPU models and four types of inference tasks. The results show that *on-the-fly adaptive speculation*, especially the online window size optimization, can deliver similar or even better improvements than the prior method that uses extensive ahead-of-time trainings, leading to 3.55-16.48% speed improvement over the standard speculative decoding, and 1.2-3.4$\times$ over the default LLMs. As a drop-in solution, this new approach needs no model changes, ahead-of-time preparation, lengthy training, or extensive benchmarking. It automatically adapts the optimal window size and directs the requests to the appropriate draft models for speculation, especially suitable for large LLM service providers.

It is worth mentioning that besides adapting the speculation process, there are some other methods explored in recent studies to improve speculative decoding (Li et al., 2024; Yan et al., 2024; Spector & Re, 2023; Hooper et al., 2023). Online Speculative Decoding (Liu et al., 2023), for instance, uses knowledge distillation to continuously train the smaller draft model during inference, enhancing performance. SpecInfer (Miao et al., 2023) introduces a tree-based decoding algorithm that uses the draft model to speculate multiple possible token sequences in parallel and then validates each of these sequences by the target model to keep the longest validated one. The *on-the-fly adaptive speculation* proposed in this current paper is from a different angle. It is hence complementary to those studies in the sense that it can be integrated into the speculation process in those solutions to further improve their effectiveness.

## 2 BACKGROUND

This section covers the key concepts, details, and techniques related to LLMs and speculations.

### 2.1 LARGE LANGUAGE MODELS

Large language models (LLMs) are natural language processing (NLP) models built on transformer architecture. Due to their large capacity, LLMs can handle a wide range of tasks, including translation, code generation, and question answering. Prompt engineering involves providing the LLM with tailored instructions and guidance to improve the quality of task-specific outputs.

For an LLM workflow, training is one-time, and it can take several months to complete; the model is then deployed for inference, which typically requires continuous serving, and often, on a large scale (e.g., over 100 million users weekly). When serving LLM inference in real-time, it is mostly memory-bound. This means that it is bounded by the memory bandwidth of the hardware it is

running on. More details about the memory-bound nature of LLM inference can be found in Appendix A. The cost of inference exceeds the cost of training over time.

## 2.2 GUESS-AND-VERIFY IN LLMS

In LLM inference, the tokens generated later are dependent on the tokens generated earlier. This sequential dependency of autoregressive decoding in LLMs has led to the development of new techniques aimed at parallelizing the decoding process. Given that text is tokenized, some tokens can be easier or harder to predict by a lower-parameter LLM. This has sparked a new area of research known as "guess-and-verify" optimization (Li et al., 2024; Yan et al., 2024; Spector & Re, 2023; Hooper et al., 2023). In this approach, smaller draft models efficiently guess a number of tokens, which are then verified in parallel by a larger target model. It is a lossless optimization, maintaining the accuracy of the results.

Speculative decoding is one typical "guess-and-verify" approach in LLM optimization. In this technique, when an LLM samples logits, it essentially predicts the probabilities of the next token. Speculative decoding takes advantage of this by allowing a smaller model to guess the easier tokens based on its own sampling of the distribution. These tokens are then verified against a larger, more accurate model.

In speculative decoding, the process involves guessing a set of tokens from the smaller model $M_q$ within a fixed window size, $\gamma$, and then verifying these $\gamma$ tokens against the larger model $M_p$ by sampling $\gamma + 1$ tokens in parallel. If all tokens are accepted, the $\gamma + 1$ tokens are appended to the generated sequence, and the process continues. If one token (say $(i+1)th$) is rejected, the algorithm accepts the $i$ correct tokens, resample the $(i + 1)th$ from an adjusted distribution in the validation, and continues. The speculation and verification process is detailed in Algorithm 1.

---

**Algorithm 1** Speculative Decoding (Leviathan et al., 2023)

---

1: **function** speculativeDecoding($M_p$, $M_q$, $prefix$)
2:     Sample $y$ guesses $x_1, \cdots, x_y$ from $M_q$ autoregressively.
3:     **for** $i = 1$ **to** $y$ **do**
4:         $q_i(x) \sim M_q(prefix + [x_1, \cdots, x_{i-1}])$
5:         $x_i \sim q_i(x)$
6:     Run $M_p$ in parallel.
7:     $(p_1(x), \cdots, p_y(x)) \leftarrow$
8:         $M_p(prefix), \cdots, M_p(prefix + [x_1, \cdots, x_{y-1}])$
9:     Determine the number of accepted guesses $n$.
10:     $r_1 \sim U(0, 1), \cdots, r_y \sim U(0, 1)$
11:     $n \leftarrow \min(\{i | 1 \le i \le y, r_i > p_i(x)\} \cap \{y\})$
12:     Adjust the distribution from $M_p$ if needed.
13:     $p'(x) \leftarrow p_{n+1}(x)$
14:     **if** $n < y$ **then**
15:         $p'(x) \leftarrow \mathcal{N}(\max(0, p_{n+1}(x) - q_{n+1}(x)))$
16:     Return one token from $M_p$ and $n$ tokens from $M_q$.
17:     $t \sim p'(x)$
18:     **return** $prefix + [x_1, \cdots, x_n, t]$

---

## 3 OVERVIEW

Speculative decoding can provide significant speedups. Current speculative decoding methods, however, use fixed model pairs and a static $\gamma$ determined for a dataset after benchmarking. The motivation of this work is thus the need to choose an appropriate draft model and an optimal $\gamma$ before deployment. Different LLM tasks and domains have varying optimal model pairings and $\gamma$ values, making a static choice suboptimal. For example, suppose we adjust the best $\gamma$ for each prompt instead of the whole dataset. In that case, we see a 9–18% increase in speedups, indicating the extra potential for additional performance if we adaptively choose a more optimal $\gamma$ at a fine-grained level.

**Overview.** We propose *on-the-fly adaptive speculation*, overcoming the deployment challenges and making it efficient and productive. The workflow goes as follows. At the beginning, it sets up the

Figure 1: Our on-the-fly adaptive speculation framework. When a prompt arrives, our scheduler directs it to the draft model $M_q$. During speculation, our framework automatically adapts the right speculation window size $\gamma$. The speculation is then validated by the target model $M_p$.

target model and different draft model options. For each prompt, our solution as in Figure 1 involves two steps. First, it finds a proper draft model for the given prompt. This is done by extracting features of the prompt to estimate the single token accuracy. From there, the method approximates the acceptance rate and ultimately the throughput so it can choose a proper draft model. Second, it runs speculations, where $\gamma$ is adapted on the fly with the given model pairing. In the following content, we will first introduce the adaptive window size selection (Section 4) followed by adaptive draft model selection (Section 5).

# 4  ADAPTIVE WINDOW SIZE SELECTION

Suppose we have fixed the target and draft model pair. In this section, we first introduce the analytic model for capturing the relationship between the speculation setting and speculation benefits. With that, we present four agile algorithms for adaptively changing $\gamma$ during speculative decoding. The agility of these algorithms is essential for minimizing the runtime overhead.

## 4.1  ONLINE WINDOW SIZE OPTIMIZATION

A speculation window size that is too large risks high overhead if verification fails early, while a size that is too small misses out on the full benefits. The optimal size varies depending on the language model, contexts, and speculation accuracy. We translate this trade-off into an objective function to adaptively determine the optimal window size across various configurations. For each prompt, we want to minimize the end-to-end latency in generating a response with a fixed number of tokens. We define our objective function as the expected number of tokens verified as correct per unit time, aiming to maximize this function by optimizing the window size $\gamma$:

**Definition 1** (formulating objective). *Let $a_q$ represent the latency of generating one token by the draft model, and $b_p(\gamma)$ represent the latency of a verification step with window size $\gamma$. For $t = 1, 2, \cdots$, let $Acc(x_t|X_{<t})$ be the accuracy of the speculation of a single token given the current context $X_{<t} = \{x_1, \cdots, x_{t-1}\}$. The window size $\gamma$ for the current speculation step can be determined by optimizing the objective*

$$\mathcal{G} = \max_{\gamma} \frac{1 - Acc(x_t|X_{<t})^{\gamma+1}}{(1 - Acc(x_t|X_{<t}))(\gamma a_q + b_p(\gamma))}. \tag{1}$$

The derivation for the objective is included in Appendix B.1.

**Algorithm.** To adaptively determine the optimal $\gamma$ using estimation for $a_q, b_p(\gamma), Acc(x_t|X_{<t})$, the algorithm goes as follows. At the start of each speculation step, it conducts the following two operations before it can solve the objective (1). First, it estimates $a$ and $b$. These values are derived by observing the most recent steps. Second, it estimates $Acc(x_t|X_{<t})$ based on the recent history. We use maximum likelihood estimation over the last $h$ speculations, ensuring the estimate $\widehat{Acc}$ reflects both locality and reduced variance (details in Appendix B.2). In our algorithm, we let $\gamma(j)$ be the speculation window size during the $j$-th most recent verification step, and $V(\gamma(j), X_{<t_j})$ the

number of accepted tokens in this speculation window. We estimate $Acc(x_t|X_{<t})$ as

$$\widehat{Acc}(x_t|X_{<t}) = \frac{\sum_j V(\gamma(j), X_{<t_j})}{\sum_j V(\gamma(j), X_{<t_j}) + \sum_j \mathbf{1}(V(\gamma(j), X_{<t_j}) < \gamma(j))} \tag{2}$$

where $\mathbf{1}(\cdot)$ is the indicator function. To avoid overly optimistic estimates and potential division-by-zero error when $\widehat{Acc}$ approaches 1, we set a fixed upper limit, $Acc_{\max}$, and cap $\widehat{Acc}$ at this value.

**Analysis.** We now analyze the throughput of the adaptive speculation.

**Theorem 1.** *Let $L$ be the length of the answer to a prompt and is fixed, $n$ be the total number of speculation steps, and $\{\gamma_q^i\}_{i=1}^n$ denote the history of the window sizes during the adaptive speculation. Let $d_q = \mathbb{E}_{i=1,\cdots,n}(\gamma_q^i)$ be the average window size during speculation, and acceptance rate $\rho_q$ be the number of accepted tokens divided by the total number of tokens sampled by $M_q$. The throughput ($R$) can be formulated as*

$$R = \frac{L}{b_p(\gamma)n + a_q \frac{L}{\rho_q}}. \tag{3}$$

*Proof.* In the following formulations, we omit $p$ and $q$ as the formulations are about a given $p$ and $q$ pair. The throughput $R$ is computed by dividing the length of the answer by the latency $t$:

$$R = \frac{L}{t}. \tag{4}$$

The total latency of generating outputs for one prompt is computed as

$$t = \sum_{i=1}^n a\gamma^i + b(\gamma) = b(\gamma)n + a\sum_{i=1}^n \gamma^i = n(b(\gamma) + a \cdot \mathbb{E}(\gamma^i)). \tag{5}$$

Inspecting the relations among $d, n, \rho$ and $L$ gives us

$$L = d \cdot n \cdot \rho. \tag{6}$$

Solving for Equations 4, 5 and 6 gives us the expression for throughput. $\qquad\square$

Figure 3 shows our adaptive window size method actually increases the acceptance rate $\rho$, and is eventually translated to improvement to throughput.

## 4.2 OTHER DROP-IN SPECULATION METHODS

Besides the online optimization method, we have explored three other methods for on-the-fly $\gamma$ adaption.

**Finite State Machine (FSM)-Based Speculation.** A finite state machine-based predictor (Hennessy & Patterson, 2017) is similar to an $n$-bit saturating counter used in branch prediction. The mechanism works by decreasing $\gamma$ by 1 if a token from the draft model is rejected, and increasing $\gamma$ by 1 if all tokens are accepted. During benchmarking, We still select a value for $\gamma$, but it is considered an upper limit, $\gamma_{\max}$. If the draft and target models' distributions significantly differ, $\gamma$ will remain low, potentially even at 0. Conversely, if the models align closely, $\gamma$ should increase, approaching $\gamma_{\max}$. We consider this approach particularly effective for natural language processing because certain parts of a sentence—like common phrases or syntactically predictable structures—are easier for a smaller draft model to predict. In contrast, more unique or complex sub-sequences might be harder to guess. By adaptively changing $\gamma$ based on the previous token validations, we create a reward system that exploits patterns and predictable structures in autoregressive generation.

**Cache-Enabled FSM-Based Speculation.** We adjust $\gamma$ based on the context provided by the prompt and the history of generated tokens. In settings like question-answering, an LLM often reiterates or directly responds based on the context given by the user. Therefore, the user's input can inform predictions about the type of response the LLM will generate. Specifically, this approach includes a token cache that updates after every sampling step. Initially, the cache is populated with tokens in the

prompt, set up before the prefill stage. As new tokens are sampled and validated during speculation, the cache is updated with any previously unseen tokens. $\gamma$ is then adjusted dynamically: It increases by one if a validated token is already in the cache, and by an additional one when all speculated tokens are accepted. Conversely, if none of the accepted tokens are in the cache, it decreases by one. We see that this approach is particularly effective for structured tasks like QA chatbot interactions or code completion, where context and history play a significant role. However, it may be less effective for short prompts expecting broad and diverse content, such as tasks that require informative or creative responses. In such cases, the lack of initial context or history means the cache is less informative, making $\gamma$ adjustments less effective, potentially leading to performance similar to the more simplistic state-based adaptation.

**Reinforcement Learning-Based Speculation.** We in addition explored a reinforcement learning (RL) based approach. We use a Q-learning agent to choose a $\gamma$. The modification to the algorithm is detailed in Algorithm 2 in Appendix B.4. The agent takes the previous states of $\gamma$ as inputs and applies an action after each validation step.

## 5 ADAPTIVE DRAFT MODEL SELECTION

Besides the speculation window size, the selection of the draft model also makes a difference: A smaller draft model can make faster inferences but at the risk of a low acceptance rate, while a larger draft model renders a longer latency.

As answer length $L$ and target model latency $b$ in Equation 4 are considered constant in our setting, the main influence for choosing a draft model comes from draft model latency $a$, the acceptance rate $\rho$, and the number of speculation steps $n$.

**Influence of selecting a larger draft model.** Let $c$ represent the inference latency ratio between the draft model and the target model. Choosing a larger draft model increases the single token accuracy, $\alpha = \mathbb{E}(Acc(x_t|X_{<t}))$, and the draft latency $a$. We estimate $d = \mathbb{E}(\gamma)$ by finding the numerical integer solution in objective 1. With $\alpha$ and the corresponding $d$, the acceptance rate $\rho = \frac{1-\alpha^{\gamma+1}}{(1-\alpha)d}$ can be determined, as shown by scattered dots in Figure 2.

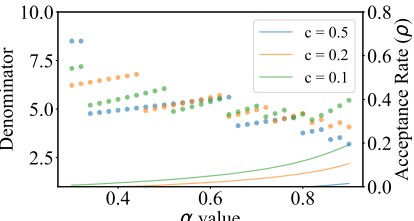

Figure 2: Results for different $\alpha$ and $c$ values.

To analyze the influence on the number of speculation steps $n$, we look into Equation 6, where $n = \frac{L}{d \cdot \rho}$. The lines in Figure 2 illustrate how the denominator of $n$ changes as $\alpha$ varies, reflecting the product of $d$ and $\rho$.

**Theorem 2.** *Let $\Delta n$ represent the reduction in speculation steps due to a larger draft model, $\Delta c$ the increase in latency ratio, and $\Delta \rho$ the improvement in acceptance rate. As long as the following condition holds:*

$$\Delta n > \frac{\Delta c}{\Delta \rho} L, \tag{7}$$

*the larger draft model would lead to a higher overall throughput than the smaller draft model.*

A deeper look into formula 7 gives us that, when comparing two draft models, $\Delta c$ can be easily determined using sample profiling results. If we are able to approximate the increase in $\alpha$, $\Delta \rho$ and $\Delta n(\alpha, d)$ can also be determined because their relation to $d$, $\alpha$, and $c$ is deterministic. Therefore, to select a suitable draft model when a new prompt arrives, we need to approximate $\alpha$ and inspect whether condition 7 holds in order to determine whether to use a larger draft model.

Typically a prompt can be represented as a vector. We represent a prompt as a vector $\mathbf{u} \in \mathbb{R}^r$ with $r > 0$ being the vector length. We model our goal $\alpha_c^{\mathbf{u}}$ of prompt $\mathbf{u}$ for a certain ratio $c$ as

$$\alpha_c^{\mathbf{u}} = \mathbf{u}^\top \mathbf{Z}_c + \epsilon_c^{\mathbf{u}} \tag{8}$$

where $\mathbf{Z}_c$ is the parameter vector to determine and the random noise variable $\epsilon_c^{\mathbf{u}}$ is independent of $\mathbf{Z}_c$. For each $\mathbf{u} \in \mathbb{R}^r$, the random variables $\{\epsilon_c^{\mathbf{u}}\}$ are identically distributed with $\mathbb{E}(\epsilon_c^{\mathbf{u}}) = 0$ for all $\mathbf{u}$. The vector embedding is constructed as a concatenation of the following features.

**Prompt Features.** We transform a prompt into a feature vector based on these features.

- *Prompt length.* Prompt length may affect the choice of draft prompts. Questions with short, open-ended prompts tend to be easier, for which, smaller draft models may suffice and offer faster speeds. Longer prompts tend to be more complex and more challenging for small draft models. $\rho$ tends to increase when a larger draft model is used for a prompt. $n$ can decrease to an extent greater than $\frac{\Delta c}{\Delta \rho} L$ which may justify the choice of a larger draft model.
- *Prompt perplexity.* The perplexity of a prompt influences the choice of draft model as well: In the case where the language model is more confident about a prompt (i.e., exhibits lower perplexity regarding its semantics), a smaller draft model is likely to generate high-quality answers with a higher $\alpha$. Perplexity here is defined as

$$\text{PPL} = 2^{-\frac{1}{N} \log_2 \Pr(w_i | W_{<i})} \tag{9}$$

  where $N$ is the length of the prompt and $\Pr(w_i | W_{<i})$ is the probability of $w_i$ given the context $\{w_1, \cdots, w_{i-1}\}$.
- *Term Frequency-Inverse Document Frequency (TF-IDF).* The other features that also affect how challenging a prompt is to LLM are the TF-IDF feature, which is computed based on the term frequency in the prompt and the term importance in the entire corpus.

**Algorithm.** Based on the analysis, we devise the following algorithm to select draft model. Suppose there exist $r$ linearly independent prompts $\mathbf{b}_1, \cdots, \mathbf{b}_r \in \mathbb{R}^r$. In the beginning, for each ratio $c$ and these $r$ prompts, the algorithm runs the speculative decoding and observes the single token accuracy $\alpha_c^{\mathbf{b}_p}$ and computes the ordinary least square estimate for $\mathbf{Z}_c$, given by

$$\widehat{\mathbf{Z}}_c = \left( \sum_{p=1}^{r} \mathbf{b}_p \mathbf{b}_p^\top \right)^{-1} \sum_{p=1}^{r} \mathbf{b}_p \alpha^{\mathbf{b}_p}.$$

For each newly arrived prompt $\mathbf{u}$, it computes the estimated $\hat{\alpha}_c^{\mathbf{u}}$ for potential draft-target model pairs and check Equation 7 to select the optimal draft model. In an LLM server center setting that has many machines hosting many LLMs, the selection of draft models can be implemented by redirecting requests to the appropriate nodes in the center equipped with the desired draft model and target model pair.

# 6 EVALUATION

In this section, we present and analyze the experimental results gathered from testing our proposed algorithms and hypotheses.

## 6.1 EXPERIMENTAL SETUPS

This part outlines the configurations and setups used to collect the performance data.

**Software.** We primarily use the HuggingFace Transformers library with PyTorch implementations. The flexibility of Python and the availability of pre-trained weights on HuggingFace allow us to experiment with various methods and conduct detailed analyses. The GPU implementations utilize NVIDIA's cuDNN library, which is optimized for large language models (LLMs) and transformers. To ensure the best performance and compatibility with the latest models, we use the most recent versions of Transformers (v4.38.2) and PyTorch (v2.2.1).

**Datasets and Models.** we used three datasets to evaluate model performance and benchmark various implementations. These datasets were selected to reflect common tasks found in chatbot settings and other LLM applications. We employed system prompts to guide the LLMs for higher-quality outputs, particularly for tasks like coding and text summarization. See Appendix C.1 for more details. The datasets include OpenAI's HumanEval (Chen et al., 2021a) for coding tasks, XSum for extreme text summarization (Chen et al., 2021b), GSM8K (Cobbe et al., 2021) for mathematical reasoning, and Alpaca (Taori et al., 2023) for complex advice queries. We include llama-2-chat 70B, Meta OPT 13B, BigScience BLOOM 7B, and Dolly 12B for target models. More details about the models we benchmarked are in Appendix C.1. Each dataset was sampled with 25 prompts in online predictive model construction, and evaluated with all remaining prompts across various settings. Note that when using speculative decoding, the draft model and the target model should

have been trained on the same datasets to achieve good prediction accuracies, which limits the possible combinations in our experiments.

**Hardware.** LLMs demand significant GPU computing power and memory, particularly during inference, where memory bandwidth is critical in achieving high throughput on GPUs. Table 1 lists the GPUs we used, their memory bandwidth, capacity, and the datatypes employed.

Table 1: GPU Hardware

| GPU | HBM (GB) | Mem Bandwidth (GB/s) | Datatype |
|---|---|---|---|
| NVIDIA V100 | 32 | 900 | FP16 |
| NVIDIA A100 | 80 | 1555 | BF16 |
| NVIDIA RTX4090 | 24 | 1008 | BF16 |

We use two NVIDIA A100 GPUs with 80G memory for the LLaMA 70B-7B pair and 70B-13B pair. We distribute the 70B model across two GPUs, which leads to communication overhead during inference with LLaMA 70B. However, for speculative decoding, the 7B (13B) draft model is only loaded on a single GPU, reducing this overhead. For other model pairs, we limit our study to one GPU, loading both the target and draft models on a single device. This approach serves two purposes. First, it allows us to explore the effects of resource constraints on a single GPU, which is relevant for future work on speculative decoding for personal devices. Second, it maximizes efficiency, as splitting a small LLM onto one GPU and a large LLM onto another would underutilize the resources; it is more effective to run both models on a single GPU.

## 6.2 PERFORMANCE

We list in Table 2 the throughput results of adaptive window size selection for different model pairs on different hardware. The results of the online window optimization method are reported. We have the following observations. First, our method achieves an average of $2.07\times$ speedups compared to original autoregressive decoding, and an additional $7.69\%$ improvement over speculative decoding baselines. Second, our method achieves different speedups when benchmarking on different datasets. For the HumanEval dataset, speculative decoding has the potential to significantly accelerate performance due to the structured nature of programming languages, which follow stricter grammar and syntax compared to natural language. Repetitive patterns, such as `for` loops or `if-else` statements, are easier for the draft model to predict accurately. With adaptive speculation, the algorithm can adjust the parameter $\gamma$ dynamically to suit different sub-sequences. For instance, $\gamma$ can be increased for predictable loops, whereas for more complex or less frequent constructs like API calls or high-level programming, $\gamma$ can be reduced to improve the alignment between the draft and target models, minimizing token waste. Third, the ratio of parameters matters when it comes to model pairing. Larger ratios generally lead to higher speedups while smaller target-draft parameter ratios such as BLOOM 7B-1B1 leave smaller room for improvement. This can be due to the very small model not providing enough throughput that overcome the large draft model.

Next, we show the results of the draft model selection. For a certain target model, when a new prompt comes in, we first decide on the draft model with the proper model size. This decision is made online. Table 3 compares the speedups over the speculative decoding with and without draft model selection. For LLaMA 70B, the draft model currently includes LLaMA 7B and LLama 13B. For BLOOM 7B, the draft model includes BLOOM 560M, 1B1, and 1B7. The overall throughput speedups from $3.55\%$ to $16.48\%$ using adaptive draft model selection.

We compare our online adaptive window size selection with SpecDec++ (Huang et al., 2024) in Table 4. SpecDec++ uses a ResNet to determine whether to stop speculation during speculative sampling at the current word predicted from the draft model. It employs this method based on its prediction of whether the next draft token will be accepted. Training this ResNet model requires conducting offline profiling runs and collecting data on the hardware (for example, 500 hours on A100-80G GPUs for training dataset generation, 400 hours for training, and 500 hours for evaluation set). The experimental setups are the same as in its paper. In the experiments, we use LLaMA-2-chat models (Touvron et al., 2023b), selecting the 7B version as the draft model and the 70B version as the target model for the A100 platform and BigScience BLOOM 560m version as the draft model and the 7B version as the target model for GTX 4090. To optimize memory usage, the

Table 2: Evaluation of adaptive window size selection. SPS denotes the throughput improvement our method achieves over the original speculative decoding. ARS denotes improvements over the default LLMs without speculative decoding. ("-" for not-runnable cases due to memory limit)

| Model Pairing | Dataset | A100 | | V100 | | 4090 | |
|---|---|---|---|---|---|---|---|
| | | SPS | ARS | SPS | ARS | SPS | ARS |
| LLaMA 70B/7B | finance-alpaca | 6.43% | 2.11× | - | - | - | - |
| LLaMA 70B/13B | finance-alpaca | 4.89% | 1.90× | - | - | - | - |
| BLOOM 7B/560M | finance-alpaca | 4.28% | 1.05× | 7.69% | 1.15× | 3.70% | 1.22× |
| BLOOM 7B/1B1 | finance-alpaca | 4.36% | 1.04× | 3.20% | 1.15× | 3.29% | 1.17× |
| OPT 13B/125M | finance-alpaca | 4.82% | 2.32× | 3.41% | 3.4× | - | - |
| Dolly 12B/3B | finance-alpaca | 9.11% | 1.03× | - | - | - | - |
| LLaMA 70B/7B | humaneval | 10.35% | 2.41× | - | - | - | - |
| LLaMA 70B/13B | humaneval | 8.53% | 2.23× | - | - | - | - |
| BLOOM 7B/560M | humaneval | 8.14% | 1.04× | 2.51% | 1.09× | 3.09% | 1.25× |
| BLOOM 7B/1B1 | humaneval | 4.03% | 1.1× | 3.57% | 1.16× | 3.51% | 1.3× |
| OPT 13B/125M | humaneval | 11.40% | 2.29× | 2.15% | 3.34× | - | - |
| Dolly 12B/3B | humaneval | 15.20% | 1.07× | - | - | - | - |
| LLaMA 70B/7B | gsm8k | 7.13% | 2.28× | - | - | - | - |
| LLaMA 70B/13B | gsm8k | 9.66% | 2.08× | - | - | - | - |
| BLOOM 7B/560M | gsm8k | 15.03% | 1.× | 2.52% | 1.01× | 4.84% | 1.18× |
| BLOOM 7B/1B1 | gsm8k | 10.70% | 1.× | 0.77% | 1.02× | 1.97% | 1.19× |
| OPT 13B/125M | gsm8k | 5.95% | 2.24× | 10.52% | 3.36× | - | - |
| Dolly 12B/3B | gsm8k | 16.92% | 1.06× | - | - | - | - |
| LLaMA 70B/7B | xsum | 2.94% | 1.73× | - | - | - | - |
| LLaMA 70B/13B | xsum | 0.14% | 1.5× | - | - | - | - |
| BLOOM 7B/560M | xsum | 77.50% | 1.× | 49.30% | 1.× | 54.63% | 1.× |
| BLOOM 7B/1B1 | xsum | 70.91% | 1.× | 42.94% | 1.× | 54.17% | 1.× |
| OPT 13B/125M | xsum | 10.64% | 1.02× | 7.91% | 2.43× | - | - |

Table 3: Throughput performance improvement over speculative decoding.

| Target Model | finance-alpaca | | | humaneval | | | gsm8k | | |
|---|---|---|---|---|---|---|---|---|---|
| | A100 | V100 | 4090 | A100 | V100 | 4090 | A100 | V100 | 4090 |
| LLaMA 70B (w/o draft selection) | 6.43% | - | - | 10.35% | - | - | 9.66% | - | - |
| LLaMA 70B (w/ draft selection) | 6.46% | - | - | 11.11% | - | - | 9.66% | - | - |
| BLOOM 7B (w/o draft selection) | 4.36% | 7.69% | 3.70% | 8.14% | 3.57% | 3.51% | 9.76% | 2.52% | 4.84% |
| BLOOM 7B (w draft selection) | 4.94% | 16.48% | 8.15% | 8.57% | 4.96% | 4.17% | 9.76% | 3.55% | 6.83% |

models are implemented in the bfloat16 format. The tok/s speedups comparison is as follows on both the A100 and 4090 devices. We find that although our method uses no ahead-of-time training while SpecDec++ uses hundreds of GPU-hours to do that, our method outperforms SpecDec++ consistently, with an average of $5.7\%$ improvement in latency. Part of the time savings come from selecting the $\gamma$ value before each speculation instead of running a neural network each time the draft model produces a new token. Our approach further shows advancement by adaptively choosing $\gamma$ on the fly without arduous data collecting and training.

Table 4: Comparison of Tok/s speedups (v.s. autoregressive) and productivity of SpecDec++ and our method (without draft model selection).

| Dataset | A100 (LLaMA 70B/7B) | | 4090 (BLOOM 7B/560m) | |
|---|---|---|---|---|
| | SpecDec++ | Ours | SpecDec++ | Ours |
| Alpaca | 2.04× | 2.11× | 1.21× | 1.26× |
| HumanEval | 2.23× | 2.41× | 1.22× | 1.23× |
| GSM8K | 2.26× | 2.28× | 1.17× | 1.18× |
| Profiling & Preperation | 1000h | 0 | 100h | 0 |
| Offline Training | 400h | | 400h | |

## 6.3 Detailed Analysis

We compare the throughput and acceptance rate for different adaptive speculation methods in Figure 3. $\gamma$ denotes the speculation window size for the original speculative decoding method and upper-bound speculation (simply by skipping the validation process); we set a maximum $\gamma_{\max}$ value for other adaptive speculation methods so $\gamma$ will be truncated if it increases over $\gamma_{\max}$. All experiments are conducted on the A100 machine with OPT 13B-125M model pair. From the figure, we find that (i) the online window size optimization method gives the best overall performance. (ii) Even though RL-based speculation gives better acceptance rates than the other methods, it shows lower throughput. This is because a higher acceptance rate is not directly linked to a higher throughput as in Equation 3. In our case, RL-based speculation remains at a low $\gamma$ value to keep the acceptance rate high while also losing the potential for more speedups. (iii) cache-based and state-based speculation perform better when prompts are longer (e.g., the humaneval dataset). This can be attributed to a more stable $\gamma$ prediction as more information is involved in the long prompt.

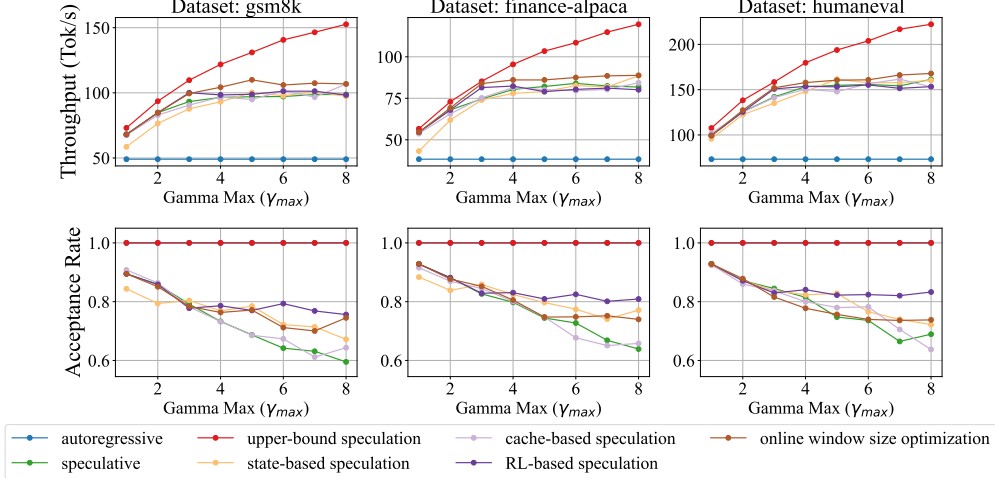

Figure 3: Detailed experimental results of different adaptive methods.

## 6.4 Results for Scalability

**Comprehensive Chat Dataset.** We include evaluations for a comprehensive chat dataset ShareGPT (Community, 2023) in Appendix C.3. Results show that our method achieves an average of $1.71\times$ speedups compared to original autoregressive decoding, and an additional $4.9\%$ improvement over speculative decoding baselines.

**Adaptive Speculation for Tree-based Decoding Method.** Current speculative decoding uses tree-based methods (Cai et al., 2024; Li et al.). The *on-the-fly adaptation of speculative decoding* is complementary to the tree-based decoding. By adaptively changing the draft tree depth, our drop-in method can adjust the draft token sequence length on the fly and thus improving decoding performance. We apply our method to the state-of-the-art EAGLE-2 (Li et al., 2024) and report the results in Appendix C.4. On the MT-Bench (Zheng et al., 2023), we achieve up to $3.56\times$ speedups compared to original autoregressive decoding, and an additional $4.2\%$ improvement over SOTA.

## 7 Conclusion

In this paper, we propose *on-the-fly adaptation for speculative decoding* to accelerate LLM inferences. As a pure software approach, it introduces a two-level adaptation for draft model adaptation and online window size adaptation with no ahead-of-time profiling or training, providing a drop-in optimization for existing LLMs. We experimentally demonstrate the effectiveness of this method and show 3.55% to 16.48% speedups compared to the speculative decoding, and $1.2\times$ to $3.4\times$ over the default LLMs without speculative decoding. Among the several online adaptive methods, we found that the token accuracy-based online window size optimization method works the best, consistently outperforming other methods in terms of the overall LLM throughput.

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

## A   MEMORY-BOUND LLM INFERENCE

When serving LLM inference in real-time, it is predominantly memory-bound, limited by the hardware's memory bandwidth. Figure 4 illustrates Meta's LLaMA 7B inference on an NVIDIA V100, where 86.7% of the runtime is spent on a General Matrix-Vector multiplication (GEMV) kernel, a common BLAS Level 2 subroutine (Blackford et al., 2002).

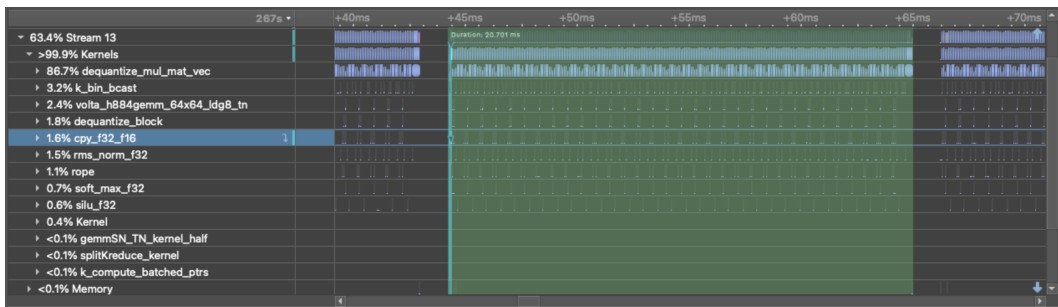

Figure 4: LLaMA 7B FP16 trace using NVIDIA NSight Systems on an NVIDIA V100 GPU with LLaMA.cpp (Han et al., 2016; NVIDIA, 2022; Gerganov, 2023). The majority of GPU time is consumed by the dequantize_mul_mat_vec function, a custom GEMV (General Matrix-Vector multiplication) operation within the library.

The breakdown highlights function calls needed to generate one token, taking nearly 20ms. Generating multiple tokens scales linearly, and BLAS Level 2 routines, being memory-bound, underutilize the compute units, making memory bandwidth the primary bottleneck in reducing latency.

## B   METHOD DETAILS

### B.1   FORMULATION OF OBJECTIVE 1

This section discusses details on formulating objective 1.

**Expected Accepted Token Length.** Given the single token accuracy $\beta = Acc(x_t|X_{<t}) \in [0,1]$, the expected accepted number of tokens is computed as:

$$
\begin{aligned}
\mathbb{E}(\text{\# of accepted tokens}|X_{<t}) &= 1 + \sum_{i=1}^{\gamma-1} i\beta^i(1-\beta) + \gamma\beta^\gamma \\
&= 1 + \sum_{i=1}^{\gamma-1} i\beta^i - \sum_{i=2}^{\gamma}(i-1)\beta^i + \gamma\beta^\gamma \\
&= \sum_{i=0}^{\gamma} \beta^i \\
&= \frac{1-\beta^{\gamma+1}}{1-\beta}.
\end{aligned}
\tag{10}
$$

**Formulation of objective.** The expected number of verified tokens as correct in a $\gamma$-long speculation window is $\frac{1-Acc(x_t|X_{<t})^{\gamma+1}}{1-Acc(x_t|X_{<t})}$. The total latency of one speculation step and verification step is calculated as $\gamma a_q + b_p$. Therefore, the expected number of tokens verified as correct per unit time given a window size $\gamma$ is

$$
\frac{1-Acc(x_t|X_{<t})^{\gamma+1}}{(1-Acc(x_t|X_{<t}))(\gamma a_q + b_p)}.
$$

## B.2 Estimation of $Acc(x_t|X_{<t})$

Let $\beta = Acc(x_t|X_{<t})$. Let $Y$ be a random variable of the number of accepted tokens truncated at $\gamma + 1$. The probability function of $Y$ is

$$f(y) = \begin{cases} \dfrac{(1-\beta)\beta^{y-1}}{1-\beta^{\gamma+1}}, & y = 1, 2, 3, \cdots, \gamma+1 \\ 0, & \text{otherwise.} \end{cases} \tag{11}$$

**Maximum Likelihood Estimation.** For a random sample of size $n$, the likelihood function is

$$L = (1 - \beta^{\gamma+1})^{-n}(1-\beta)^n \beta^{\sum_{i=1}^n y - n}.$$

The following equation 12, a $d$th-degree polynomial in $\hat{\beta}$, provides the maximum likelihood estimator for $\beta$.

$$\left(\sum_{i=1}^n y_i - n(\gamma+2) + n\right)\hat{\beta}^{\gamma+2} + \left(n(\gamma+2) - \sum_{i=1}^n y_i\right)\hat{\beta}^{\gamma+1} - \left(\sum_{i=1}^n y_i\right)\hat{\beta} + \sum_{i=1}^n y_i - n = 0 \tag{12}$$

Given values of $\gamma$, $n$, and $\sum_{i=1}^n y_i = n\overline{y}$, one can compute the value of $\hat{\beta}$ using an iterative technique such as the Newton-Rhapson method to solve equation 12. It can be shown that there is only one root in the range $0 < \hat{\beta} < 1$.

To eliminate the need for an iterative solution to equation 12, we maintain a table to provide approximate solutions. From equation 12,

$$\overline{y} = \{\gamma\hat{\beta}^{\gamma+2} - (\gamma+2)\hat{\beta}^{\gamma+1} + 1\}/(\hat{\beta}^{\gamma+2} - \hat{\beta}^{\gamma+1} - \hat{\beta} + 1). \tag{13}$$

Further observation gives us that the rate of change of $\overline{y}$ concerning $\hat{\beta}$ appears to be sufficiently constant, making linear interpolation feasible and enabling our approximation in equation 2.

## B.3 Optimal Gamma

Given the single token accuracy and inference latency ratio of the draft model to the target model $c$, the optimal $\gamma$ value to optimize objective 1 can be determined as in Figure 5.

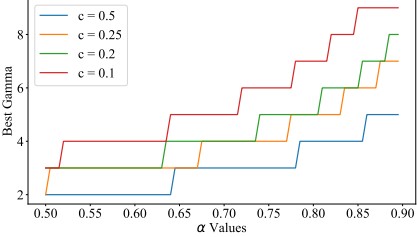

Figure 5: The optimal $\gamma$ for different $\alpha$ and $c$ values.

### B.4 PSUDO-CODE FOR REINFORCEMENT LEARNING-BASED SPECULATION

---

**Algorithm 2** Reinforcement Learning-Based Speculative

---

1: **function** reinforcementLearningSpeculation($M_p$, $M_q$, $prefix$, Agent)
2:     Sample $y$ guesses $x_1, \cdots, x_y$ from $M_q$ autoregressively.
3:     **for** $i = 1$ **to** $y$ **do**
4:         $q_i(x) \sim M_q(prefix + [x_1, \cdots, x_{i-1}])$
5:         $x_i \sim q_i(x)$
6:     Run $M_p$ in parallel.
7:     $(p_1(x), \cdots, p_y(x)) \leftarrow$
8:         $M_p(prefix), \cdots, M_p(prefix + [x_1, \cdots, x_{y-1}])$
9:     Determine the number of accepted guesses $n$.
10:    $r_1 \sim U(0, 1), \cdots, r_y \sim U(0, 1)$
11:    $n \leftarrow \min(\{i | 1 \leq i \leq y, r_i > p_i(x)\} \cap \{y\})$
12:    Adjust the distribution from $M_p$ if needed.
13:    $p'(x) \leftarrow p_{n+1}(x)$
14:    **if** $n < y$ **then**
15:        $p'(x) \leftarrow \mathcal{N}(\max(0, p_{n+1}(x) - q_{n+1}(x)))$
16:    action $\leftarrow$ GetAction(Agent, $y$)
17:    $y$ = action
18:    Reward = the percentage of the accepted speculated tokens
19:    Return one token from $M_p$ and $n$ tokens from $M_q$.
20:    $t \sim p'(x)$
21:    **return** $prefix + [x_1, \cdots, x_n, t]$

---

## C ADDITIONAL EXPERIMENTS

This section includes additional experimental results.

### C.1 ADDITIONAL EXPERIMENTAL SETUPS

**Prompt Dataset.** Table 5 consolidates information on the datasets, tasks, and additional details we used to benchmark and compare performance.

Table 5: Prompt Dataset

| Dataset | Task | System Prompt |
|---|---|---|
| OpenAI HumanEval | Code completion | You are an expert programmer that helps to complete Python code. Given the code from the user, please complete the rest of the code to the best of your ability. |
| XSum | Summarization | You write two sentence summaries of new articles. Do not write any more. Keep it brief and to the point. |
| GSM8K | Math Word Problem | You are given a math question, and your task is to answer it. Then provide a step-by-step walkthrough on how you got the answer to the question. |
| Finance-Alpaca | Finance QA | You are a finance expert. Answer the following questions to the best of your knowledge, and explain as much as possible. |

**Models.** When implementing speculative decoding, selecting appropriate model pairs presents challenges. The parameter ratio is crucial, as a low ratio can negate speed gains if the draft model isn't significantly faster than the target model. Additionally, both models must share the same tokenizer to avoid conversion overhead from differing tokenization schemes (Schuster & Nakajima, 2012; Sennrich, 2015). Speculative decoding is more effective with models trained on similar datasets, as seen with Meta's LLaMA models (Touvron et al., 2023b;a) or DeepMind's Chinchilla. Mixed precision (FP16 or BF16) is preferred, avoiding quantization due to slowdowns, and using deterministic

decoding with a temperature of 0 for consistency (Hinton, 2015). Dolly is an open-source model from Databricks aimed at democratizing LLMs by offering open-source weights and the datasets needed for instruction fine-tuning (Conover et al., 2023). The following table 6 details the model pairs.

Table 6: Model Card

| Target Model | Draft Model | Same Vendor? | Ratio |
|---|---|---|---|
| Meta LLaMA 70B | Meta LLaMA 13B | Yes | 5.4x |
| Meta LLaMA 70B | Meta LLaMA 7B | Yes | 10x |
| BigScience BLOOM 7B | BigScience BLOOM 560M | Yes | 12.5x |
| BigScience BLOOM 7B | BigScience BLOOM 1.1B | Yes | 7x |
| Meta OPT 13B | Meta OPT 125M | Yes | 96.3x |
| DataBricks Dolly 12B | DataBricks Dolly 3B | Yes | 4.0x |

**Implementation Details.** The FSM-based method and cache-enabled FSM-based method are inspired by branch prediction in computer architecture (Lee et al., 1997; Smith, 1998; Jiménez & Lin, 2001). The reinforcement learning-based speculation involves online learning, so we conducted 25 warmup trials before recording benchmarks. To minimize overhead, the RL algorithm runs on the CPU rather than the GPU, ensuring both inference and training are completed in under 1 millisecond. This makes the overhead negligible when considering the end-to-end latencies compared to standard speculative decoding.

## C.2 ADDITIONAL EXPERIMENT RESULTS

We include more experiment results. Figure 6 and Figure 7 compare the throughput and acceptance rate for different adaptive speculation methods on the A100 machine with the BLOOM BigScience 7B-560M model pair and LLaMA 70B-7B.

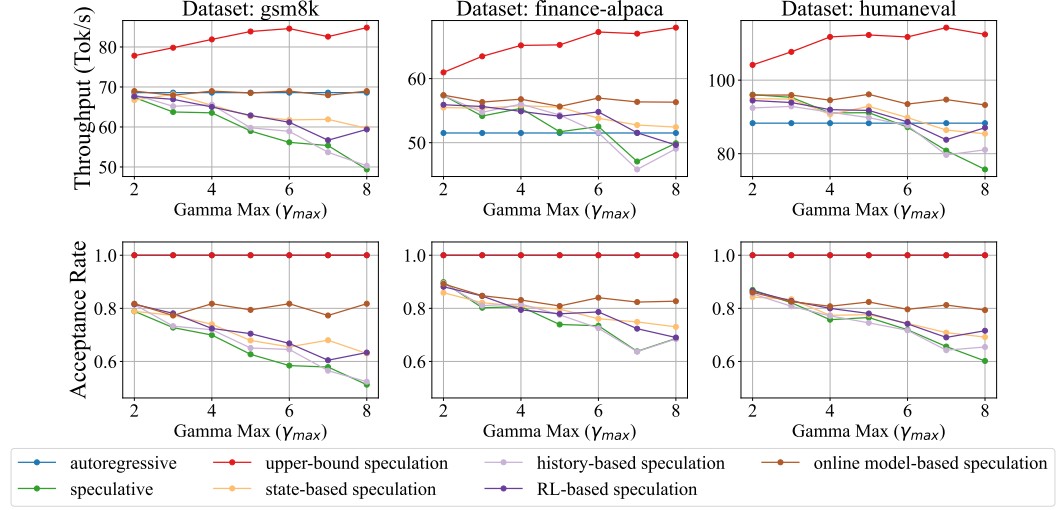

Figure 6: Detailed experimental results for BLOOM 7B-560M.

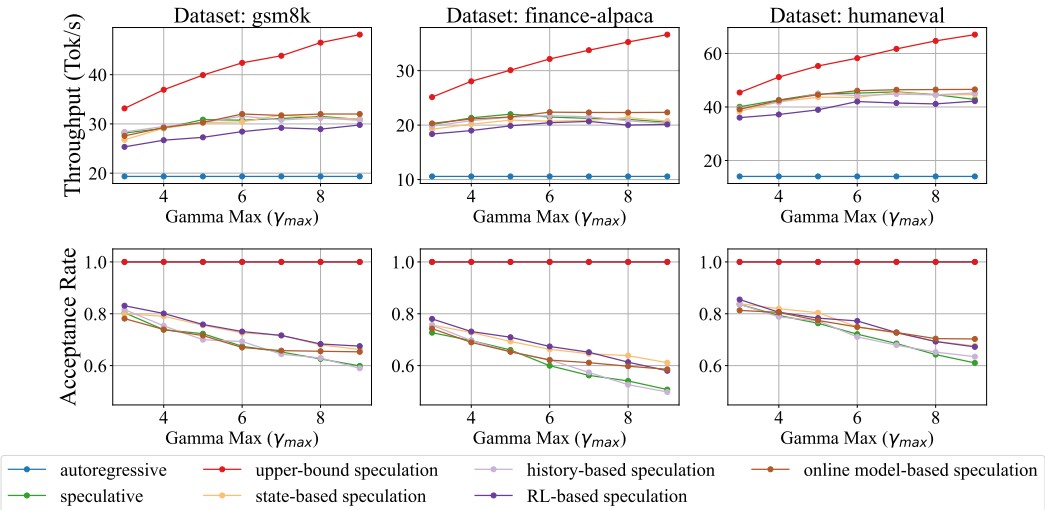

Figure 7: Detailed experimental results for LLaMA 70B-7B.

## C.3 COMPREHENSIVE CHAT DATASET

Table 7 shows the throughput results of adaptive window size selection for different model pairs on different hardware on the shareGPT dataset. The results of the online window optimization methods are reported. The experimental setups are the same as in Section 6.1.

Table 7: Evaluation for the comprehensive chat dataset. SPS denotes the throughput improvement our method achieves over the original speculative decoding. ARS denotes improvements over the default LLMs without speculative decoding.

| Hardware | Model Pairing | Dataset | Throughput | |
| --- | --- | --- | --- | --- |
| | | | SPS | ARS |
| A100 | LLaMA 70B/7B | shareGPT | 7.89% | 2.20× |
| | LLaMA 70B/13B | shareGPT | 3.69% | 1.92× |
| | OPT 13B/125M | shareGPT | 4.81% | 2.10× |
| 4090 | BLOOM 7B/560M | shareGPT | 4.58% | 1.18× |
| | BLOOM 7B/1B1 | shareGPT | 3.50% | 1.18× |

## C.4 ADAPTIVE SPECULATION FOR TREE-BASED DECODING

Table 8 shows the results of adaptive tree depth selection on EAGLE-2 for different model pairs on different hardware. The experimental setups are the same as in Section 6.1. In addition, Table 9 provides a detailed analysis of serving latency, speculation latency, verification latency, and speculation accuracy. Speculation latency is measured as the number of tokens selected from the draft tree per second. Our method shows lower speculation latency compared to EAGLE-2. This is because, while we dynamically adapt the tree depth, we keep the number of tokens selected from the draft tree the same as in EAGLE-2. However, with a larger tree depth, more tokens might sampled due to the increased number of layers. Verification latency is similar for both EAGLE-2 and our method, as they utilize the same target model. Notably, our method improves the acceptance rate by dynamically adjusting the tree depth, which effectively changes the speculation window size.

Table 8: Evaluation for adaptive speculation in improving EAGLE-2, a method for tree-based speculative decoding. SPS denotes the throughput improvement our method achieves over EAGLE-2. ARS denotes improvements over the default LLMs without speculative decoding. ("-": model is out of memory)

| Target Model | Dataset | A100 | | 4090 | |
|---|---|---|---|---|---|
| | | SPS | ARS | SPS | ARS |
| Vicuna-7B-v1.3 | MTBench | 3.44% | 2.72× | 6.22% | 2.28× |
| LLaMA2-Chat 7B | MTBench | 2.66% | 3.13× | 6.23% | 2.72× |
| LLaMA2-Chat 13B | MTBench | 2.22% | 2.56× | - | - |
| LLaMA2-Chat 70B | MTBench | 1.46% | 3.56× | - | - |
| LLaMA3-Inst 70B | MTBench | 1.14% | 2.68× | - | - |

Table 9: Detailed analysis for adaptive speculation in improving EAGLE-2. "Speculation" and "Verification" denote speculation latency and verification latency, respectively. (Unit for latency: Toks/sec)

| Hardware | Target Model | Method | Serving Latency | Speculation | Verification | Acceptance Rate |
|---|---|---|---|---|---|---|
| A100 | Vicuna-7B-v1.3 | EAGLE-2 | 82.44 | 472.58 | 708.31 | 0.67 |
| | | Ours | 85.27 | 446.71 | 666.01 | 0.72 |
| | LLaMA2-Chat 7B | EAGLE-2 | 97.81 | 569.05 | 4491.42 | 0.62 |
| | | Ours | 100.41 | 299.85 | 4591.73 | 0.66 |
| | LLaMA2-Chat 13B | EAGLE-2 | 79.74 | 558.02 | 4535.35 | 0.61 |
| | | Ours | 81.51 | 491.37 | 4530.73 | 0.62 |
| | LLaMA2-Chat 70B | EAGLE-2 | 27.50 | 389.38 | 4532.27 | 0.51 |
| | | Ours | 27.90 | 192.02 | 4491.19 | 0.65 |
| | LLaMA3-Inst 70B | EAGLE-2 | 24.33 | 266.33 | 3392.65 | 0.53 |
| | | Ours | 24.61 | 132.08 | 3300.60 | 0.65 |
| 4090 | Vicuna-7B-v1.3 | EAGLE-2 | 117.95 | 665.97 | 1041.83 | 0.56 |
| | | Ours | 125.28 | 579.34 | 1164.03 | 0.56 |
| | LLaMA2-Chat 7B | EAGLE-2 | 142.15 | 712.72 | 8278.28 | 0.67 |
| | | Ours | 151.00 | 643.91 | 8137.47 | 0.72 |

## C.5 SENSITIVITY STUDY

**Effects of different history length.** Table 10 shows a sensitivity study for the effects of different history lengths when adjusting the window size. The results are collected on the A100 machine for the BLOOM 7B-560M pair.

Table 10: Sensitivity study for different history length values when adjusting window size. The best throughput is highlighted for each $\gamma_{max}$.

| Dataset | History Length | $\gamma_{max}$ | | | |
|---|---|---|---|---|---|
| | | 5 | 6 | 7 | 8 |
| Alpaca | 5 | 52.28 | 52.49 | 52.05 | 52.37 |
| | 6 | **54.18** | **53.46** | 52.71 | 53.00 |
| | 7 | 53.03 | 53.01 | **54.32** | **53.36** |
| Humaneval | 5 | 93.98 | 94.48 | **94.21** | **94.21** |
| | 6 | **94.84** | **95.18** | 93.39 | 93.39 |
| | 7 | 94.54 | 94.30 | 93.41 | 93.41 |
| gsm8k | 5 | 62.69 | **62.15** | 63.42 | 64.03 |
| | 6 | 61.48 | 61.78 | **63.72** | 61.84 |
| | 7 | **64.77** | 61.38 | 62.74 | **64.27** |

**Effects of vector length.** Table shows the sensitivity study for the effects of different vector dimensions for model selection. The results are collected on the 4090 machine for the BLOOM 7B-560M pair.

Table 11: Sensitivity study for different dimensions for model selection.

| Dimension | 4 | 8 | 10 | 12 | 16 |
|---|---|---|---|---|---|
| Throughput | 74.29 | 75.23 | 74.00 | 75.46 | **75.55** |

