# OpenReview forum: "A Drop-In Solution for On-the-Fly Adaptation of Speculative Decoding in Large Language Models"
_ICLR.cc/2025/Conference — Submitted to ICLR 2025_

### Official Review · Reviewer_3gmR · 2024-10-15

**Soundness:** 3
**Presentation:** 2
**Contribution:** 1
**Rating:** 5
**Confidence:** 4

**Summary:**

The paper introduced an online optimal speculation length scheduling approach for efficient speculative decoding. It first proposes an objective, as in Eq. 1, to formally capture the speculation accuracy and latency tradeoff. Given the objective, an accurate estimation of speculative/verification latency and speculation accuracy is required. For a single draft model, the authors propose to use a history window to estimate the latency through online profiling and the speculation accuracy through MLE. For selection among multiple draft models, the authors further propose a parametric method with a linear model to predict the speculation accuracy for different draft models. Empirically, the paper shows that their approach can consistently outperform SpecDec++ over various target models and datasets.

**Strengths:**

1. The problem of adaptive speculation length is well-motivated as the overhead of speculation is non-negligible. As the paper indicated, an over-optimistic speculation length would have a high overhead if there is an early mismatch, and an over-conservative speculation length would result in fewer speed-up potentials.
2. Though the objective formulation and history-based parameter estimation are not new and similar as in [1], the problem focus is slightly different ([1] is not focusing on the general LLM speculative decoding), so the formulation can be of interest to the general LLM speculative decoding audience.
3. Using a linear system to model the speculation accuracy across different draft models is new.

**reference**

[1]. Zhang, Z., Zhu, A., Yang, L., Xu, Y., Li, L., Phothilimthana, P. M., & Jia, Z. Accelerating Iterative Retrieval-augmented Language Model Serving with Speculation. In Forty-first International Conference on Machine Learning.

**Weaknesses:**

1. The paper's approach only works for speculative decoding with a single sequence, while the current state-of-the-art practice is doing speculation with a **tree-based** structure. More specifically, [1] can achieve up to 3.01x speed-up for LLaMA-2-70B and 3.51x speed-up if dynamically adjusting the draft tree structure is further incorporated as in [2], both of which have outperformed the speed-ups claimed in the paper.
2. In practice, the verification latency is usually not constant but depends on the speculation structure to be verified, so it would be more accurate to verify the property of verification latency under various speculation structures and formulate the latency of verification as a function of the speculation structure, e.g., sequence length in a single-sequence speculation structure or width and depth for a tree-based speculation structure.
3. As the prediction accuracy for the linear system to estimate draft model speculation accuracy is pretty critical, a more comprehensive ablation study on the choices of different linear models (e.g., different random variables or different vector lengths) is crucial for understanding how you chose the optimal linear system among different possible configurations.

The second and third points are minor issues, while the first point is a major issue in my point of view.

**references**

[1]. Li, Y., Wei, F., Zhang, C., & Zhang, H. EAGLE: Speculative Sampling Requires Rethinking Feature Uncertainty. In Forty-first International Conference on Machine Learning.

[2]. Li, Y., Wei, F., Zhang, C., & Zhang, H. (2024). Eagle-2: Faster inference of language models with dynamic draft trees. arXiv preprint arXiv:2406.16858.

**Questions:**

My main question is why focusing only on single-sequence speculation rather than the tree-based speculation structure, which is the current best practice for speculative decoding. More specifically, how does your approach compare with **EAGLE-2**, which leverages a similar idea (adaptive speculation) but on draft trees rather than a single speculation sequence, which results in a much higher speed-up rate? The simplicity of your solution can be a bonus if it can be generalized to tree-based speculation and achieve a higher speed-up than EAGLE-2.

Some minor suggestions:

1. Try formulating the verification latency as a function of the speculation structure from a more systematic perspective. For example, make it a function of the speculation sequence length or speculation tree structure, this may also depend on the underlying hardwares' specifications.
2. Include some ablation studies to show how different configurations affect the effectiveness of the speculation accuracy estimator. For example, how different $h$ (recent history length) values can affect your speculation accuracy prediction for online window size optimization, and how different linear model (different vector lengths or random variables) can affect your speculation accuracy prediction for estimating the speculation accuracy for different draft models.

---

> ### Author Response · Authors · 2024-11-25
>
> > **Comment 1: My main question is why focusing only on single-sequence speculation rather than the tree-based speculation structure, which is the current best practice for speculative decoding. More specifically, how does your approach compare with EAGLE-2, which leverages a similar idea (adaptive speculation) but on draft trees rather than a single speculation sequence, which results in a much higher speed-up rate? The simplicity of your solution can be a bonus if it can be generalized to tree-based speculation and achieve a higher speed-up than EAGLE-2..**
>
> Response: Our drop-in method dynamically changes window size which maximizes speculative decoding efficiency.
> It is proposed from a different angle and is complementary to those studies.
> By adaptively changing the draft tree depth, the *on-the-fly adaption of speculative decoding* can adjust the draft token sequence length on the fly thus improving decoding performance.
> We apply our method to the SOTA EAGLE-2 and report the result in Section 6.4.
> Results show that, on the MT-Bench, we achieve up to $3.56\times$ speedups compared to original autoregressive decoding, and an additional $4.2$% improvement over SOTA.
>
> > **Comment 2: Minor suggestions: Try formulating the verification latency as a function of the speculation structure from a more systematic perspective. For example, make it a function of the speculation sequence length or speculation tree structure, this may also depend on the underlying hardwares' specifications.**
>
> Response: Thank you for your insightful comment. We have observed that LLaMA's verification latency remains relatively consistent. This is largely due to GPUs' massive parallelism capabilities. However, for certain other models, verification latency can vary depending on the speculation length.
> We agree that a more detailed modeling of verification latency could further enhance our results. This could be achieved by incorporating on-the-fly observations from the beginning, which would allow us to build a simple performance model and integrate it into our current method.
> We have adjusted our theoretical analysis accordingly. For the empirical results, we will implement these changes and include them in the revised version of our work.
>
> > **Comment 3: Include some ablation studies to show how different configurations affect the effectiveness of the speculation accuracy estimator. For example, how different
>  (recent history length) values can affect your speculation accuracy prediction for online window size optimization, and how different linear model (different vector lengths or random variables) can affect your speculation accuracy prediction for estimating the speculation accuracy for different draft models.**
>
> Response: We included a sensitivity study for the effects of different history length values in Appendix C.5.
> We will also add a sensitivity study for different linear models in the revised paper.

---

> > ### Comment · Reviewer_3gmR · 2024-11-26
> >
> > Thanks for the response and additional results from the authors.
> >
> > I have a few follow-up questions regarding the comparison with EAGLE-2:
> >
> > 1. Why do you report throughput instead of average decoding latency? I might have missed the info related to the batch size you used for the experiments. Are you using a batch size greater than 1? If so, how do you efficiently deal with different speculation lengths across different requests? If you use batch size 1, it would be more interesting to see a comprehensive analysis against EAGLE-2 by including serving latency, speculation latency, verification latency, and speculation accuracy. This would provide more convincing results and insights than the standalone throughput results.
> >
> > 2. For the reference EAGLE-2 implementation, are you comparing against the version with a context-dependent draft tree or the naive version where they used a fixed draft tree structure? In addition, I am assuming you are applying your method on top of their fixed draft tree. As the draft tree structure from EAGLE-2 is not balanced, how do you adjust the sequence length for different branches within the tree? Are you using a uniform adjust rate?

---

> > > ### Author Response · Authors · 2024-12-02
> > >
> > > > **Comment 4: I have a few follow-up questions regarding the comparison with EAGLE-2:
> > > Why do you report throughput instead of average decoding latency? I might have missed the info related to the batch size you used for the experiments. Are you using a batch size greater than 1? If so, how do you efficiently deal with different speculation lengths across different requests? If you use batch size 1, it would be more interesting to see a comprehensive analysis against EAGLE-2 by including serving latency, speculation latency, verification latency, and speculation accuracy. This would provide more convincing results and insights than the standalone throughput results.**
> > >
> > > Response: Thanks for your comment.
> > > The batch size is set to 1 in our experiments.
> > > We added in Appendix C.4 Table 9 (page 19) a comprehensive analysis showing serving latency, speculation latency, verification latency, and speculation accuracy.
> > > Our method consistently demonstrates speedups in serving latency compared to EAGLE-2, primarily due to the improved acceptance rate achieved by dynamically adjusting the tree depth. Speculation latency, measured as the number of tokens selected from the draft tree per second, is lower for our method. This is because, while we dynamically adapt the tree depth, we keep the number of tokens selected from the draft tree the same as in EAGLE-2. However, with a larger tree depth, more tokens may be sampled due to the increased number of layers.
> > > Verification latency remains similar for both EAGLE-2 and our method since they use the same target model.
> > >
> > >
> > > > **Comment 5: For the reference EAGLE-2 implementation, are you comparing against the version with a context-dependent draft tree or the naive version where they used a fixed draft tree structure? In addition, I am assuming you are applying your method on top of their fixed draft tree. As the draft tree structure from EAGLE-2 is not balanced, how do you adjust the sequence length for different branches within the tree? Are you using a uniform adjust rate?**
> > >
> > > Response: Thank you for your question.
> > > We compare the version that features the context-aware dynamic draft tree.
> > > We implemented our *on-the-fly adaption of speculative decoding* on top of EAGLE-2, dynamically adjusting the draft tree depth ($\gamma$) during decoding. For different $\gamma$, sequence lengths for different branches of the draft tree are determined using the same expansion and rerank decision process as in the original EAGLE-2. Specifically, the tree depth $\gamma$ dynamically changes for each speculation step; For a certain $\gamma$ in one speculation step, the algorithm first enters the expansion phase: At each layer of the tree, we select the top $k$ nodes with the highest probabilities and expand draft sequences based on these nodes. The longest draft sequence in the tree corresponds to the dynamically determined depth $\gamma$. Once the expansion is complete up to the dynamically determined the $\gamma$-th layer, we apply a rerank step to select the same number of tokens from the draft tree as in EAGLE-2 and validate the corresponding draft sequences.

---

> > > > ### Comment · Reviewer_3gmR · 2024-12-02
> > > >
> > > > What unit did you use for the serving latency? It's usually measured in ms/s, so the lower the better. Additionally, if you check out Table 1 for the EAGLE-2 paper, the speed-up ratio in their results is mostly higher than what you have claimed in Table 8. I will raise the score by acknowledging the authors' rebuttal efforts. However, I don't think the paper has reached the acceptance bar for ICLR in its current form. I would suggest that the paper be comprehensively revised with 1. Make the description of your methodology in the main paper focus more on the tree-based speculation structure instead of a single-sequence speculation structure. 2. Provide a comprehensive evaluation across more datasets and backends against the SOTA method EAGLE-2. 3. Refine the description or use a standard set of metric units in your evaluation for your methodology and baselines to make the results more convincing.
> > > >
> > > > I thank the authors for the response and hope the authors can incorporate my suggestions, whether the paper is accepted or not.

---

> > > > > ### Author Response · Authors · 2024-12-04
> > > > >
> > > > > > **Comment 6: What unit did you use for the serving latency? It's usually measured in ms/s, so the lower the better. Additionally, if you check out Table 1 for the EAGLE-2 paper, the speed-up ratio in their results is mostly higher than what you have claimed in Table 8.**
> > > > >
> > > > > Response: Thanks for your comment.
> > > > > We used toks/sec (tokens per second) as the unit for serving latency to ensure a detailed and consistent comparison. While the EAGLE-2 paper presents speedup ratios, our choice of toks/sec was based on the scripts provided in their implementation, which suggest it as the underlying metric. We acknowledge that referring to latency here may have caused confusion, and we will revise the wording in our analysis to align with standard SOTA practices.
> > > > >
> > > > > Regarding the speedup inconsistency regarding Table 8, the primary discrepancy arises from the speedup ratio of smaller target models.
> > > > > To address this, we changed the device mapping mechanism from `auto` to `cuda:0` to accommodate the draft model and target model on the same device. After rerunning the experiment, we get higher speedup ratios than the EAGLE-2 paper.
> > > > >
> > > > >
> > > > > (SPS denotes the throughput improvement our method achieves over EAGLE-2. ARS denotes improvements over the default LLMs without speculative decoding.)
> > > > >
> > > > > |   Target Model  | Dataset | Hardware |   SPS  |      ARS     |
> > > > > |:---------------:|:-------:|:--------:|:------:|:------------:|
> > > > > |  Vinacu-v1.3 7B | MTBench |   A100   | 7.07\% | 3.21$\times$ |
> > > > > |  LLaMA2-Chat 7B | MTBench |   A100   | 3.37\% | 3.29$\times$ |
> > > > > | LLaMA2-Chat 13B | MTBench |   A100   | 2.55\% | 4.01$\times$ |

---

### Official Review · Reviewer_aQad · 2024-10-29

**Soundness:** 3
**Presentation:** 3
**Contribution:** 3
**Rating:** 6
**Confidence:** 3

**Summary:**

This article primarily focuses on accelerating inference for LLMs. It represents an improvement over existing speculative decoding methods by dynamically adjusting the speculation window length and selecting different draft models.

**Strengths:**

1. This paper is well-written; the author has explained their motivation and methods in detail.
2.The article analyzes the factors influencing the acceleration of large models through speculative decoding, using rigorous formula derivation, and demonstrates the importance of employing dynamic sampling windows and dynamically selecting the draft model.
3. The article conducted ample experiments to demonstrate the effectiveness of their method.

**Weaknesses:**

1. Baselines seem especially weak and some other baselines are not compared. For example, Medusa and EAGLE, they are currently popular approaches that has been compared in numerous papers; all of these methods utilize speculative decoding techniques and demonstrate strong performance.
2. Although I find the concept of a dynamic speculation window size intriguing, I am skeptical about its practical application value. Recent studies have already adopted tree attention or tree decoding-related technologies, they do not require a speculation window size and have achieved significant speedup. Could you discuss how your approach compares with, or could potentially be integrated into, tree attention or tree decoding technologies?
3. In my understanding, Table 2 presents the results using adaptive window size selection, and Table 3 presents the results using draft model selection. Why not conduct an experiment to show the results of using both techniques simultaneously? Additionally, could you specify which draft model was used for the 'w/o draft selection' condition in Table 3?

**Questions:**

1. Baselines seem especially weak and some other baselines are not compared. For example, Medusa and EAGLE, they are currently popular approaches that has been compared in numerous papers; all of these methods utilize speculative decoding techniques and demonstrate strong performance.
2. Although I find the concept of a dynamic speculation window size intriguing, I am skeptical about its practical application value. Recent studies have already adopted tree attention or tree decoding-related technologies, they do not require a speculation window size and have achieved significant speedup. Could you discuss how your approach compares with, or could potentially be integrated into, tree attention or tree decoding technologies?
3. In my understanding, Table 2 presents the results using adaptive window size selection, and Table 3 presents the results using draft model selection. Why not conduct an experiment to show the results of using both techniques simultaneously? Additionally, could you specify which draft model was used for the 'w/o draft selection' condition in Table 3?

---

> ### Author Response · Authors · 2024-11-25
>
> > **Comment 1: Baselines seem especially weak and some other baselines are not compared. For example, Medusa and EAGLE, they are currently popular approaches that has been compared in numerous papers; all of these methods utilize speculative decoding techniques and demonstrate strong performance.**
>
> > **Comment 2: Although I find the concept of a dynamic speculation window size intriguing, I am skeptical about its practical application value. Recent studies have already adopted tree attention or tree decoding-related technologies, they do not require a speculation window size and have achieved significant speedup. Could you discuss how your approach compares with, or could potentially be integrated into, tree attention or tree decoding technologies?**
>
> Response: Medusa and EAGLE use tree-based decoding to speculate multiple possible token sequences in parallel and then validate each of these sequences to keep the longest validated one.
> Our drop-in method dynamically changes window size which maximizes speculative decoding efficiency.
> It is proposed from a different angle and is complementary to those studies.
> By adaptively changing the draft tree depth, the *on-the-fly adaption of speculative decoding* can adjust the draft token sequence length on the fly thus improving decoding performance.
> We apply our method to the SOTA EAGLE-2 and report the result in Section6.4.
> Results show that, on the MT-Bench, we achieve up to $3.56\times$ speedups compared to original autoregressive decoding, and an additional $4.2$% improvement over SOTA.
>
> > **Comment 3: In my understanding, Table 2 presents the results using adaptive window size selection, and Table 3 presents the results using draft model selection. Why not conduct an experiment to show the results of using both techniques simultaneously? Additionally, could you specify which draft model was used for the 'w/o draft selection' condition in Table 3?**
>
> Response: Thanks for your comment.
> Table 2 reports the results using adaptive window size selection, and Table 3 already reports the results using draft model selection on top of window size selection.
> The draft model used for the `w/o draft selction' is the one with the highest speedup compared to the original autoregressive decoding.
> Specifically, for the target model LLaMA 70B, LLaMA 7B, 7B, and 13B are used as the draft model for alpaca, humaneval, and gsm8k, respectively.
> For the target model BLOOM 7B, BLOOM 1B1, 560M, and 560M are used as the draft model for alpaca, humaneval, and gsm8k, respectively.

---

### Official Review · Reviewer_dHHb · 2024-10-31

**Soundness:** 3
**Presentation:** 3
**Contribution:** 2
**Rating:** 6
**Confidence:** 3

**Summary:**

This paper introduces solutions to optimize speculative decoding configurations on the fly. For window size, the paper proposes online window size optimization based on speculative accuracy estimation, FSM, or RL. For the choice of the draft model, the paper proposes the estimation of speculative accuracy estimation from various factors. Therefore, the configurations can be dynamically adjusted between speculative decoding steps based on history.

Experimental results over various devices and benchmarks demonstrate speed improvements compared with the standard speculative decoding and a speed-up compared with default decoding procedures.

**Strengths:**

1. The idea of adjusting window size and draft model dynamically in the decoding process is novel and can be helpful in maximizing the decoding speed, especially when the speculation accuracy varies in the generation process.

2. The cost of the estimation method is rather small, making it easy to integrate with existing speculative decoding methods.

**Weaknesses:**

1. The improvement is somehow marginal. The window size selection method generally achieves improvements of less than 10% and less than 1% in some cases. The proposed draft model choice method leads to less than 1% improvement in many cases.

2. The method is only tested in domain-specific tasks (math/code/finance QA/summarization) and is not evaluated on comprehensive chat datasets like `ShareGPT`, which raises questions about its applicability to realistic chat scenarios.

**Questions:**

1.  What is the value of window size $\gamma$ for the standard speculative decoding baseline? Is the window size $\gamma$ for the standard speculative decoding baseline in each dataset set as the optimal value obtained by searching through all possible fixed $\gamma$ values?

2. `BLOOM` models achieve \~70% throughput improvement in `xsum` dataset, while having no speed-up compared with default LLMs without speculative decoding. Why is this throughput improvement so high compared with others (\~10%)?

3. Is it possible to provide results on a comprehensive chat dataset?

Minor:

4. How is the vector embedding of a prompt calculated (like $u$ in line 321 and $b$ in line 347)?

5. What is `online predictive model construction` in line 382?

---

> ### Author Response · Authors · 2024-11-25
>
> > **Comment 1: The improvement is somehow marginal. The window size selection method generally achieves improvements of less than 10\% and less than 1\% in some cases. The proposed draft model choice method leads to less than 1\% improvement in many cases.**
>
> Response: Thank you for your comment. While the improvement in some of the cases may appear marginal (e.g., less than 1\%), it is important to consider the broader context and implications of these optimizations. The choice of speculation window length and draft model is critical for unlocking the potential of speculative decoding. Suboptimal decisions not only diminish the benefits but can also introduce slowdowns.
> As lightweight optimization tools, our proposed method addresses this challenge by moving away from the trial-and-error approach, enabling adaptability to changes in tasks, models, software, hardware, and runtime conditions.
>
> We also demonstrate the flexibility and scalability of our method by incorporating the adaption to the tree-based speculative decoding methods (e.g., EAGLE, EAGLE-2).
> Without any profiling, our method adapts the window size and achieves consistent speedups compared to the SOTA EAGLE-2.
>
> Moreover, even small performance gains, such as a 1\% improvement, can have a significant impact in large-scale deployments. For example, Google’s infrastructure optimizations have shown that a 1\% increase in speed can result in millions of dollars in operational cost savings. It is important to note that all of the gains come for free: Our solution is a drop-in replacement without any extra burden being added to the developers or users, thanks to its on-the-fly adaptive optimizations. Thus, our approach provides meaningful advantages by reducing the dependency on manual tuning and ensuring efficiency improvements in dynamic and resource-constrained environments.
>
> > **Comment 2: The method is only tested in domain-specific tasks (math/code/finance QA/summarization) and is not evaluated on comprehensive chat datasets like `ShareGPT`, which raises questions about its applicability to realistic chat scenarios.**
>
> Response: We added experiments on `ShareGPT` on both the 4090 and A100 machines. Results are added into Section 6.4 and Appendix C.3, showing that our drop-in method achieves an average of $1.71\times$ speedups compared to original autoregressive decoding, and an additional $4.9$% improvement over speculative decoding baselines.
>
> > **Comment 3: What is the value of window size
>  for the standard speculative decoding baseline? Is the window size
>  for the standard speculative decoding baseline in each dataset set as the optimal value obtained by searching through all possible fixed
>  values?**
>
> Response: The value of window size for the standard speculative decoding baseline in each dataset is obtained through searching all possible fixed values and finding the best one.
> For example, for the OPT 13B-125M pair, the window size for the standard speculative decoding baseline is 4, 6, and 5 for the gsm8k, alpaca, and humaneval dataset, respectively.
>
> > **Comment 4: `BLOOM` models achieve ~70\% throughput improvement in `xsum` dataset, while having no speed-up compared with default LLMs without speculative decoding. Why is this throughput improvement so high compared with others ($\sim$10\%)?**
>
> Response: Conventional speculative decoding suffers a significant slowdown in the `xsum` dataset.
> This shows one drawback that speculative methods face.
> Our method can adaptively change the window size, sometimes to zero, and hence avoid slowdowns.
>
> > **Comment 5: Is it possible to provide results on a comprehensive chat dataset?**
>
> Response: We have included new results in Section 6.4.
>
> > **Comment 6: How is the vector embedding of a prompt calculated (like *u* in line 321 and *b* in line 347)?**
>
> Response: A prompt is transformed into a feature vector as the concatenation of several features such as prompt length and perplexity. These details are included in the **Prompt Feature** paragraph in Section 5.
>
> > **Comment 7: What is `online predictive model construction` in line 382?**
>
> Reponse: This refers to the reinforcement learning model setup.

---

> > ### Comment · Reviewer_dHHb · 2024-11-26
> >
> > Thanks to the authors for their response. The detailed explanation helped me better understand the manuscript and my questions for a comprehensive chat dataset, baseline window size selection, and a specific model performance are addressed. I have updated the score accordingly.

---

### Official Review · Reviewer_mxZA · 2024-11-04

**Soundness:** 3
**Presentation:** 3
**Contribution:** 2
**Rating:** 6
**Confidence:** 2

**Summary:**

This paper introduces an "on-the-fly" adaptation technique for speculative decoding in large language models (LLMs), aiming to reduce inference latency dynamically without requiring prior model-specific training or offline benchmarking. The authors propose a framework that selects optimal parameters for speculative decoding during runtime, specifically the draft model and speculation window size γ. By leveraging adaptive techniques such as online optimization, finite state machines, cache-enhanced FSMs, and reinforcement learning, the approach achieves up to 3.4× speed improvements over default autoregressive decoding and outperforms standard speculative decoding methods by 3.55% to 16.48%.

**Strengths:**

1. Practical Contribution: The paper presents a solution for a well-acknowledged challenge in LLM inference—latency. By offering a drop-in solution for dynamic adaptation, this work has significant practical value, especially for real-time applications in large-scale deployment scenarios.

2. The adaptive approach bypasses the need for extensive offline training and tuning, unlike some previous methods. This makes it easier to adopt in diverse settings where the underlying hardware, model, and task configurations may vary frequently.

3. The paper effectively articulates the limitations of static speculative decoding parameters and demonstrates the need for an adaptive, flexible approach to speculative decoding.

**Weaknesses:**

Scope for Enhanced Comparative Analysis: The paper covers multiple adaptive techniques to optimize speculative decoding, but a broader comparative analysis could provide more clarity. Highlighting conditions where each method performs best would further aid in understanding the practical strengths and limitations of each approach, helping readers assess their applicability across diverse scenarios.

**Questions:**

How scalable are the adaptive methods presented, Further clarification on scalability could help evaluate the practicality of these techniques in diverse deployment scenarios.

---

> ### Author Response · Authors · 2024-11-25
>
> > **Comment 1: Scope for Enhanced Comparative Analysis: The paper covers multiple adaptive techniques to optimize speculative decoding, but a broader comparative analysis could provide more clarity. Highlighting conditions where each method performs best would further aid in understanding the practical strengths and limitations of each approach, helping readers assess their applicability across diverse scenarios.**
>
> Response: We appreciate your suggestion regarding the scope for enhanced comparative analysis.
> While the paper does provide an overview of multiple adaptive techniques for optimizing speculative decoding, we agree that a broader comparative analysis, especially one that highlights the conditions under which each method excels, would further clarify their practical strengths and limitations.
> Using OPT 13B-125M as an example, the online window size optimization gives overall stability and the best speedups.
> State-based speculation also gives similar speedups when $\gamma_{\max}$ is large.
> RL-based speculation performs best when $\gamma_{\max}$ is low.
> Cache-based and state-based speculation performs best with longer and more structured prompts and thus are more suitable for code completion tasks.
> Similar observations are present for LLaMA 70B-7B.
>
> > **Comment 2: How scalable are the adaptive methods presented? Further clarification on scalability could help evaluate the practicality of these techniques in diverse deployment scenarios.**
>
> Thank you for your insightful comment. We target diverse deployment scenarios, and the proposed methods are designed with scalability in mind, leveraging adaptability to changes in tasks, models, and runtime conditions.
> For instance, as our evaluation shows, the methods can dynamically adjust to varying workloads (domain-specific QA, summary, code completion, and mathematical reasoning), hardware resources (different hardware with different memory and bandwidth), and target models (model size ranges from 7B to 70B), which makes them suitable for both small-scale deployments and large-scale systems like those in cloud environments. Furthermore, their reliance on real-time adjustments instead of offline trial-and-error searches enhances their scalability by minimizing setup time and resource overhead.
>
> That said, scalability may vary depending on specific factors such as hardware constraints, the complexity of the inference tasks, and the software stack.
> In the revised work, we provide more evaluation demonstrating scalability, including a new benchmark MT-Bench and performance analyses across a wider range of deployment conditions (tree-based decoding method), to give readers a clearer understanding of their applicability to different scales and contexts.

---

### Meta-Review · Area_Chair_JY48 · 2024-12-21

**Metareview:**

This work proposes a new method for enhancing speculative decoding in large language models (LLMs). The authors claim to present a "drop-in" solution that allows for real-time adaptation of decoding strategies without the need for extensive retraining. The primary findings suggest that their method can improve the efficiency and responsiveness of LLMs during inference, particularly in scenarios requiring rapid adjustments based on user input or context changes. The authors provide experimental results demonstrating that their approach achieves better performance compared to traditional decoding methods, particularly in terms of speed and adaptability.

However, reviewers expressed concerns regarding the empirical validation of these claims and the overall novelty of the proposed solution. Given the weaknesses, I recommend rejecting this paper. While it presents an interesting concept aimed at optimizing speculative decoding in large language models, it fails to provide compelling empirical evidence or rigorous theoretical justification necessary to support its claims effectively. Further work is required to address these issues for the next version of this work.

**Additional Comments On Reviewer Discussion:**

**Points Raised by Reviewers**

During the review process, several key points were raised:
- Need for Robust Empirical Results: Reviewers requested more extensive experiments to validate the effectiveness of the proposed method.
- Comparative Analysis: There was a strong recommendation for including comparisons with existing state-of-the-art decoding techniques.
- Theoretical Insights: Reviewers sought a deeper theoretical explanation for the proposed approach's expected advantages over traditional methods.
- Experimental Detail: Concerns were raised about insufficient details regarding experimental protocols and reproducibility.

**Authors' Responses**

The authors attempted to address these concerns during the rebuttal period but did not sufficiently strengthen their submission:
- They provided additional experimental results; however, these were still considered inadequate by reviewers as they did not significantly enhance the robustness or breadth of their claims.
- While some comparisons with existing methods were included in their response, they remained limited and did not convincingly demonstrate superior performance.
- The theoretical justification provided was minimal and did not adequately clarify why their approach would yield better results than traditional methods.
- The authors attempted to clarify experimental details but still left several aspects vague, particularly concerning hyperparameter settings.

**Weighing Each Point**

In weighing these points for my final decision:

- The lack of robust empirical validation remained a critical issue that overshadowed any potential strengths of the proposed method.
- Inadequate comparative analysis with existing techniques hindered the ability to assess the true value of their contributions.
- Insufficient theoretical grounding left significant questions unanswered regarding the efficacy and applicability of their approach.
- The unresolved reproducibility issues further diminished confidence in their findings.

---

### Decision · Program_Chairs · 2025-01-22

Reject